# Do you have COVID-19? How to increase the use of diagnostic and contact tracing apps

**Deborah Martínez[1], Cristina Parilli[2], Ana María Rojas[1], Carlos Scartascini [1] \*, Alberto Simpser[3]**

**1** Research Departement, Inter-American Development Bank, Washington, DC, United States of America, **2** Development Effectiveness Department, Inter-American Development Bank, Washington, DC, United States of America, **3** Political Science Department, ITAM, Mexico City, CDMX, Mexico

\* carlossc@iadb.org

## Abstract

Diagnostic and contact tracing apps are a needed weapon to contain contagion during a pandemic. We study how the content of the messages used to promote the apps influence adoption by running a survey experiment on approximately 23,000 Mexican adults. Respondents were randomly assigned to one of three different prompts, or a control condition, before stating their willingness to adopt a diagnostic app and contact tracing app. The prompt emphasizing government efforts to ensure data privacy, which has been one of the most common strategies, reduced willingness to adopt the apps by about 4 pp and 3 pp, respectively. An effective app promotion policy must understand individuals' reservations and be wary of unintended reactions to naïve reassurances.

**Data Availability Statement:** All replications files are available from Harvard's Dataverse (accession number(s) doi:10.7910/DVN/Y85AEK), available at

## Introduction

Many public policies aiming for societal benefit require that individuals undertake actions with positive external effects but private costs—real or perceived. In such cases, compliance rates will tend to fall short of the level needed to attain the policy's goals. Typical examples include vaccination (where the risks are overwhelmingly perceived rather than real), energy consumption, and water use. In the context of the current COVID-19 pandemic, the adoption of contact tracing and self-diagnostic apps is an important instance of this type of policy. Widespread adoption could effectively contain or stop the spread of the disease-causing virus, but individuals have been hesitant to download and use the apps, largely due to privacy concerns [1–5]. These fears have also affected policy-makers decisions [6, 7], as it happened in South Carolina [1] and in Norway [8].

From the policy maker's perspective, the challenge is how to motivate individuals to comply, while complying with the highest ethical standards [9–11]. A common approach is to take action to mitigate the potential risks to the individual and to reassure the public that the risks are low. However intuitive, the strategy of explicitly addressing the public's worst fears may be counterproductive insofar as it fails to credibly allay the fears and instead focuses attention on them. The present study tests this general proposition in the context of the adoption of COVID-19 diagnostic and contact tracing apps.

https://dataverse.harvard.edu/dataset.xhtml?
persistentId=doi:10.7910/DVN/Y85AEK.

**Funding:** The Inter-American Development Bank
and ITAM provided support in the form of salaries
for all the authors (IDB: DM, CP, AMR, CS; ITAM:
AS), but did not have any additional role in the
study design, data collection and analysis, decision
to publish, or preparation of the manuscript. The
specific roles of these authors are articulated in the
'author contributions' section.

**Competing interests:** The authors declare no
competing interests. The Inter-American
Development Bank and ITAM provided support in
the form of salaries for all the authors, but did not
have any additional role in the study design, data
collection and analysis, decision to publish, or
preparation of the manuscript. The specific roles of
these authors are articulated in the 'author
contributions' section. The authors affiliations does
not alter our adherence to PLOS ONE policies on
sharing data and materials as it is articulated in the
'Methods' and 'Additional information' sections.

Contact tracing is a cost effective technological tool to reduce infection rates [12, 13]. It works by notifying those who have been in contact with known virus carriers and asking them to self-isolate for a few days. More than 45 governments worldwide have launched apps that allow individuals to: (a) run a self-diagnostic, and (b) receive information about whether they have been in recent contact with an infected person [14–16]. Large technology companies including Apple and Google have added to the effort by building on their existing technology and reach within communities [17]. Despite the potential impact of the apps, very few people have downloaded them. Downloads and intentions to use have been lower than acceptance levels, highlighting an intention-behavior gap [18]. Those who have adopted them tend to be the same than those who have widely adopted other preventive measures [19]. In the US states that have adopted the technology, downloads have ranged from about 10% in Virginia to 1% in Wyoming [1], and there is no widespread support for government action encouraging everyone to download and use contact tracing apps [20]. In the case of Mexico, the country in which we conduct our research, the diagnostic app (COVID-19MX) never gained much interest from the population. In the rest of the developed world, uptake has also been limited to a minority of the population: ranging from about 40% in Iceland, 26% in Australia to less than 2% in France [21, 22]. These adoption rates fall very short of what is needed for the apps to be effective [22–25]. One of the most commonly cited concerns is data privacy, with people fearing that contact-tracing apps may be tracking their whereabouts and accumulating personal information [5]. Data protection and privacy were some of the main topics discussed by the media in Germany, Austria and Switzerland, and many articles raised questions of whether authorities could be trusted to uphold data protection and privacy [5, 26].

Governments have taken many steps to improve the working of the apps and to ensure data privacy [7]. They have also focused their advertisement campaigns on the work they have been doing to ensure privacy [27–30]. Data privacy is one of the main concerns experts show when making recommendations for increasing adoption [21]. Still, focusing too much on data privacy in the public discourse, even if that is in fact the main concern preventing people from downloading and using these apps, may backfire. There is evidence, for example, that mentioning crime, even in the context of decreasing trends can provoke a 'knee-jerk' reaction that focuses people's mind on the existence of crime instead of focusing on the trend [31]. One of the potential mechanisms behind this effect may be 'priming': subtle cues in the environment may have significant, reliable effects on behavior [32]. Priming is increasingly used to study the effects of the environment on preferences [33], including affecting attitudes that enhance gender gaps [34]. Importantly for our research, priming may affect decisions regarding download of apps [35], and priming for privacy may led to increased concern while choosing apps [36]. Kahneman (2012) [37] presents a thorough discussion regarding the replicability and power of priming studies, and [38] presents recent evidence suggesting that priming effects are real.

Importantly, that negative reaction could affect the likelihood of adoption. Seen from this perspective, the mere mention of data privacy issues may be triggering in some respondents a perception of threat that makes it less likely, not more, that they will adopt a contact tracing app. This kind of behavior may be undergirded by well known cognitive biases, including attribute substitution [39] and availability heuristics [40]. Continuous emphasis on data privacy and security—even if the goal is to reassure—may generate an overestimation of data privacy risks [41].

Following on these behavioral principles, we test whether different messages make a difference for stated willingness to adopt diagnostic and contact tracing apps. We run a survey experiment in a sample of over 23,000 individuals from Mexico recruited through Facebook advertisements and email campaigns to participate in a COVID-19 survey. We randomly

allocated respondents to four treatment conditions including a pure control and three treatment vignettes. These vignettes were designed to compare the effectiveness of a data-privacy-oriented message, similar to those governments are using, to other messages used by both the public and the private sector on the willingness to download the diagnostic and contact tracing apps. We assume that these vignettes may act on judgment and behavior by activating mental concepts through subtle cues, that is, through priming [33].

The Treatment 1 group received a vignette focusing on the role of Facebook as a tool to connect people. The Treatment 2 group was exposed to a vignette highlighting the work the Mexican government has been doing to make it possible for citizens to conduct bureaucratic procedures online rather than in person, which increases welfare. The Treatment 3 group was exposed to a vignette that emulated the adoption-promotion messages that many countries are providing their citizens: "the government is working hard to ensure data privacy protection." Treatments 1 and 2, thus, do no mention privacy concerns, while Treatment 3 does. Respondents in every group, including the control, were then asked whether they would be willing to download a COVID-19 diagnostic app, and, separately, whether they would be willing to download a contact tracing app.

We find, consistent with expectations based on behavioral research, that highlighting the fact that the government is working hard to ensure data privacy decreases the average respondent willingness to adopt by about 3 percentage points for the contact tracing app and 4 percentage points for the diagnostic app, in comparison with the control condition. In contrast, the other two treatments either had no effect (Treatment 1) or increased willingness to download the app (Treatment 2). Treatment 2, which focused on government efforts to move bureaucratic procedures online and emphasized the resulting gains in convenience, in fact increased stated willingness to download the diagnostic app by about 2 percentage points.

The results are robust to softening the data privacy message provided in Treatment 3: a message focusing on the work the government was carrying out to provide data security and stating that "the data privacy of Mexicans is a priority for the government," showed the same negative effects in a separate sample of about 1,000 Mexicans.

Our results suggest that the most obvious approach to increasing adoption—directly addressing privacy issues—may not be the best, and in fact it may have counterproductive effects. A focus on fixing data privacy issues may activate data privacy fears or, alternatively, signal that data privacy is a more important issue than one believed. A different approach—such as one that highlights the goodwill of government or the convenience of online apps—may be potentially more effective. In other words, highlighting value to the citizen, rather than risk, might be a more effective way to motivate adoption. It also helps to increase trust and highlights the role of civic virtue in public health [42]. Beyond the specific context of the COVID-19 pandemic, our results provide experimental evidence that straightforward priming can importantly influence (self-reported) behavior intentions.

## Methods

### Participant recruitment and data

We conducted a survey experiment embedded within a larger survey focusing on COVID-19 experiences, attitudes, and behaviors. The survey was approved by the IRB of the Instituto Tecnológico Autónomo de México (ITAM) on July 1, 2020, under the name "Social and Behavioral Drivers of Individual Compliance with Preventive Measures during the COVID-19 Epidemic in Mexico" (memorandum letter of approval available upon request from the authors.) In the same survey, we also included the survey experiment described and analyzed in Martínez et al., (2021) [43]. Therefore, recruitment methods and sample description are the

same for both articles. The survey experiments have been designed to be orthogonal to each other to ensure there is no cross-contamination, and randomization into treatment and controls is independent of each other. We obtained written informed consent from all participants in this study. No minors were recruited.

The questionnaire was pre-tested on a small sample of colleagues and acquaintances, and subject to the IRB's recommendations. Survey respondents were recruited through a Facebook ad campaign and a separate email campaign. The Facebook ad campaign targeted a general audience composed of individuals over 18 years of age living in the Mexican states of Sonora and Guanajuato, it was associated with the official Facebook account of the Inter-American Development Bank (IDB), and it was run by the Knowledge, Innovations and Communications Department of the IDB. The ads can be found in S1 and S2 Figs in the Supplementary Material. The ad was very simple, consisting of a photograph and a short text inviting people to share their COVID-19 related experiences. The campaign took place between July 7 and July 21, 2020. The second recruitment channel consisted of an email sent by various ministries of the Guanajuato state government in Mexico, using, their email distribution lists on Sendy. The list of ministries that participated in this recruitment process by providing their contact lists are the following: Ministry of Economic Development, Ministry of Tourism, Ministry of Health, and Ministry of Education. This email campaign consisted of two rounds of invitations that took place on July 10 and July 17, 2020 and no exclusion criteria were applied.

The Facebook ads directed respondents to a dedicated project webpage within the IDB website where respondents were able to access the baseline survey. The invitations from the government ministries did not direct respondents to the dedicated project webpage within the IDB website, but instead led respondents directly to the baseline survey. The survey was programmed in Qualtrics and could be completed either on a computer or a mobile device. The baseline survey itself stated on the welcome page that participation was voluntary and that respondents could end the survey at any time and for any reason. It also stated that only those who were at least 18 years of age should respond, even though neither the survey nor the treatments contain any age-inappropriate content. At the end of the survey, we asked respondents whether the individual recommended using her responses in our analysis or not according to how confident the person felt about the quality of her own responses. We made it clear that there were no consequences for selecting "Do not use." A total of 52,507 people clicked on the Facebook ad, yielding 15,542 complete and usable surveys. In addition, 14,059 people clicked on the email ad, yielding 7,642 complete and usable surveys. For purposes of the present study, we pooled all usable survey responses from both recruitment channels, for a total of 23,184 respondents. We separately recruited a third sample of about 1,000 respondents via a different email sent out by the Government of Sonora to their preexisting mailing list. We use this smaller sample for robustness test (S2 Table shows summary statistics for this sample).

A majority of respondents indicated that they would be willing to download the app. About 92% of respondents answered that they would probably or surely download the tracing app. The equivalent figure for the diagnostic app is 88% (S3 Fig in the Supplementary Material displays the distribution of responses for the control group). These numbers exceed the typical fraction of people who actually download these kinds of apps in countries where they are available, and they suggest that one or more of the following possibilities are at work: (i) People are not overly concerned about privacy; (ii) people feel that the diagnostic app is either more intrusive or less useful (or both) than the contact tracing app; (iii) social desirability bias is inflating the fraction of people who state that they would download either app. Note that the high fraction of people who report willingness to download the apps in the control condition creates the potential for a ceiling effect constraining the ability of treatment arms T1-T3 to increase take-up.

**Table 1. Balance table.**

| Variable | Control | Difference w.r.t. control | | | Observations |
|---|---|---|---|---|---|
| | (av. & s.e.) | T1 | T2 | T3 | |
| | (1) | (2) | (3) | (4) | (5) |
| Age | 1.417 | 0.008 | 0.016 | 0.005 | 22,896 |
| | (0.007) | (0.010) | (0.010) | (0.010) | |
| 1.Younger 25 | 0.208 | 0.000 | 0.001 | -0.005 | 22,896 |
| | (0.005) | (0.008) | (0.008) | (0.008) | |
| 1.Older 55 | 0.101 | 0.009* | 0.015** | 0.005 | 22,896 |
| | (0.004) | (0.006) | (0.006) | (0.006) | |
| 1.Female | 0.674 | -0.006 | -0.023*** | -0.007 | 23,072 |
| | (0.006) | (0.009) | (0.009) | (0.009) | |
| Education (group) | 2.600 | -0.011 | -0.005 | -0.014 | 22,925 |
| | (0.008) | (0.012) | (0.012) | (0.012) | |
| 1.College | 0.682 | -0.009 | -0.006 | -0.006 | 22,925 |
| | (0.006) | (0.009) | (0.009) | (0.009) | |
| 1.Exposed Covid | 0.653 | 0.004 | 0.000 | -0.004 | 22,806 |
| | (0.006) | (0.009) | (0.009) | (0.009) | |
| 1.Death Covid | 0.575 | 0.021** | 0.015* | -0.006 | 22,954 |
| | (0.007) | (0.009) | (0.009) | (0.009) | |
| 1.Older 65 | 0.266 | -0.014* | 0.008 | -0.007 | 23,093 |
| | (0.006) | (0.008) | (0.008) | (0.008) | |
| Pr(Infection) | 51.591 | -0.088 | -0.786 | 0.153 | 22,964 |
| | (0.379) | (0.530) | (0.531) | (0.538) | |
| Pr(Hospital) | 45.146 | 0.301 | -0.028 | 0.308 | 22,988 |
| | (0.336) | (0.470) | (0.471) | (0.478) | |
| 1.Attend Party | 0.125 | -0.006 | -0.005 | -0.000 | 23,087 |
| | (0.004) | (0.006) | (0.006) | (0.006) | |
| 1.Visit | 0.431 | -0.010 | -0.015 | -0.002 | 23,085 |
| | (0.007) | (0.009) | (0.009) | (0.009) | |
| 1.Risky Inside | 0.723 | 0.013 | 0.020** | 0.017* | 23,184 |
| | (0.006) | (0.008) | (0.008) | (0.008) | |
| 1.Social Distance | 0.361 | 0.000 | 0.008 | -0.014 | 23,098 |
| | (0.006) | (0.009) | (0.009) | (0.009) | |

*Notes*: Each row shows statistics for a different observable variable we have. Survey questions that serve the basis for the variables here, are available in S1 Appendix.
Column [1] shows the sample average and the standard deviation in parentheses for the control group. Columns [2]-[4] show the regression coefficient and the standard error in parentheses corresponding to an OLS regression. Column [5] shows the sample size for each regression. Standard errors are robust.
*** $p < 0.01$,
** $p < 0.05$,
* $p < 0.1$.
Variables *Age* and *Education* are tabulated according to ranges; as such they are categorical, with a higher category number referring to an older age and more years of education, respectively. 1.x refers to dummy variables.
*Source*: Authors' calculations.

The first column of Table 1 provides basic descriptive statistics for the control group. The average respondent is female (67%), completed secondary education (about 58% of the individuals in the sample have completed secondary education or higher), and reported knowing someone who had previously been exposed to COVID-19 (65%), and someone who has died

of COVID-19 (57%). About 12% of the sample reported having attended a party in the last 7 days, 43% reported having visited family members in the last 7 days, 72% reported that it is risky to perform activities in enclosed spaces such as gyms or restaurants, and 36% thinks that their neighbors keep social distance from others.

The population in our sample is more female and more educated than the average Mexican person as per the 2010 Mexican Population Census. For example, the proportion of females in the census is 51%. Moreover, the share of Mexicans with superior (post-secondary) or university education is about 22%, which is appreciably lower than in our sample. We cannot precisely estimate age in our sample because respondents were asked to select an age bracket. Our median respondent is in the category [25–39] and the median Mexican person is 29 years old. However, older individuals appear to be underrepresented in our sample, as 15% of the population is 55 years or older, in comparison with about 10% in our sample (by design, we do not sample minors) (Mexican census and demographic data are available from INEGI at https://www.inegi.org.mx/). The sample recruited using Facebook is similar in the age distribution and other demographic variables to the sample recruited via the state government's email list (the main difference is that the Facebook sample is closer to the population mean in the share of women answering the survey). This suggests that any self-selection into our sample is similar when recruiting via Facebook vs. government email. Both recruitment methods likely over-represent individuals who use computers and smartphones, namely the younger and more educated. However, we have no reason to suspect that this slant towards the younger and more educated affects the external validity of the results, since our recruited population likely resembles the population of those who could potentially download and use the health apps.

## Experimental design

Every individual in our sample was randomized into one of four treatment conditions, including a pure control. In the three other conditions, individuals were exposed to a priming vignette followed by a related question, which differed across the treatment conditions. Subsequently, all individuals were asked two outcome questions about their willingness to download, respectively, a COVID-19 diagnostic app and a contact tracing app. Those in the control condition were not shown a vignette/related question—they were only asked the two outcome questions.

The design of the vignettes aimed to test different approaches to promoting app adoptions, including the currently popular approach of reassuring the user about her data privacy, as well as two alternative approache. One of the alternative approaches focuses on the usefulness of the most popular social networking app, Facebook, while the second alternative approach focuses on the convenience of using online means to conduct business with government. In each vignette, the related question at the end aimed to reinforce the priming effect.

The vignette/related question in Treatment 1 (T1) specifically highlights the usefulness of Facebook as a tool to keep in touch with friends. The original Spanish text is provided in the Supplementary Material. The text reflects the spirit and tone of Facebook's own campaigns: "Facebook was built to bring people close together and build relationships" [44, 45]. The objective of this vignette was to highlight the usefulness of mobile apps and their popularity, despite their widely-publicized failure to guarantee full data privacy (for an overview of Facebook data privacy problems see [46–48]).

> **T1**: *Facebook is the most popular social networking tool in Mexico and in the world. It allows its users to share pictures, news, and personal information with their friends. In addition,*

*through its mobile app, it allows frequent contact with loved ones. Do you agree that the Face-book mobile app increases contact with your loved ones?*

[Yes/No]

The vignette/question in Treatment 2 (T2) focuses on the convenience of online services and on government efforts to move bureaucratic procedures online. This is based on actual efforts by the Mexican government aiming to: "provide information, services, and a platform for participation to the population. . .[and to revolutionize] the relationship between the citizen and the state" [49]. The Mexican government's digital strategy has also been copied and pursued by regional and local governments [50].

**T2**: *The government of Mexico has shifted many in-person bureaucratic procedures to online platforms. In addition, thanks to mobile apps, some of those procedures can be performed from any location. For example, Mexicans can now pay fines online at any time and from any location. Do you agree that online services increase the welfare of Mexicans?*

[Yes/No]

Treatment 3 (T3) highlighted government efforts to address and mitigate data privacy concerns.

**T3**: *Online platforms and mobile apps, which we can be used to make online purchases and pay for services, can have security issues. The government of Mexico is working very hard to protect data privacy so no Mexican is worried or affected by it. Is data protection an important issue for you?*

[Yes/No]

As a robustness check, we later present an alternative vignette also built around privacy concerns but using a different rhetorical structure. The goal of the robustness analysis is to ensure that it is the emphasis on data privacy concerns, and not some other idiosyncratic feature of the vignette, that is driving the effects we find.

The outcome questions were:

**Diagnostic application**: *If a federal government app were available for your smartphone that could help you to identify coronavirus symptoms, and inform you what to do, at no cost, and with no data usage, would you download it to your phone?*

[Definitely yes / I think so / I don't think so / Definitely not]

**Contact Tracing application**: *If, in addition to the previously-described features, the app could also alert you if you had been in contact for more than 15 minutes with an infected person, and it notified the people who were near you if you became infected, without identifying personal information (yours or others'), would you download the app?*

[Definitely yes / I think so / I don't think so / Definitely not]

Table 1 explores balance on covariates across the four treatment conditions, focusing only on covariates collected in the baseline survey prior to treatment assignment. The first column of the table provides means and standard deviations for those eventually assigned to the control group. The next three columns (2–4) provide the differences between that group and each

one of the other treatment groups. Only 4 out of 45 coefficients are significant at the 5 percent level or higher, and the sizes of these differences are substantively small. We take this as evidence that the randomization was successful. As mentioned previously, in addition to the main sample, we recruited an additional sample of about 1,000 respondents, which we use to test for robustness. S2 Table verifies balance on predetermined covariates across treatment arms for this sample.

## Estimation strategy

Due to randomization, average causal effects can be estimated by regressing each of the two outcome variables (respectively derived from each of the two outcome questions) on a set of treatment-condition indicators, minus an omitted reference category (the control group). We estimate the following linear regression model:

$$y_i = \alpha + \beta_1 T_{1,i} + \beta_2 T_{2,i} + \beta_3 T_{3,i} + u_i, \tag{1}$$

where $y_i$ is the value of a dependent variable (either stated willingness to download the diagnostic app or stated willingness to download the tracing app) for individual $i$. For the main analysis, we code the dependent variables as dichotomous variables taking the value of 0 for responses "definitely no" and "I don't think so" and the value of 1 for responses "I think so" and "definitely yes." Thus, Eq 1 can be interpreted as a linear probability model. We also present results of ordered logit models using the original four response categories.

The variable $T_{1,i}$ takes the value of 1 if respondent $i$ was assigned to the vignette emphasizing Facebook's usefulness to keep in contact with others, and the value of 0 otherwise; $T_{2,i}$ similarly indicates assignment to the vignette focusing on the government's efforts to shift bureaucratic procedures online; and $T_{3,i}$ indicates assignment to the vignette about the government's efforts to protect data privacy. The coefficients $\beta_n$, $n = 1, 2, 3$, respectively estimate the causal effects of treatment assignment—in comparison with assignment to the control—on the probability of answering either "I think so" or "definitely yes." These coefficients estimate intent-to-treat effects, since we do not observe whether respondents actually read or paid attention to the assigned vignette. Therefore, our estimates constitute lower bounds to treatment-on-the-treated effects.

## Results and discussion

Estimates for the analysis with the dichotomized dependent variables are shown in Table 2. Columns 1 and 5 display Eq 1 estimates with no additional controls, respectively for the contact tracing app and diagnostic app outcome questions. Columns 2–4 and 6–8 also control for a broad set of pre-treatment variables, with the goal of adjusting for potential imbalances (however small) and potentially increasing the precision of the treatment effect estimates. The set of control variables includes: age, sex, and educational attainment, whether the respondent or somebody she knows has been exposed to or has died because of COVID-19, beliefs regarding the probability of being infected and/or having to go to the hospital, whether the respondent attended a party or visited family recently, her evaluation of the risk of contagion associated with indoor activities, and beliefs about whether others around her practice social distancing. Columns 3 and 7 additionally control for state fixed effects, and columns 4 and 8 control for municipal fixed effects, instead. The results change very little across specifications. Fig 1 presents dot plots of regression coefficients corresponding to columns 2 and 6 in the table. The dot plots also display the coefficients for all the control variables.

Individuals assigned to treatment T3, which refers to government efforts to ensure data privacy, are 4 percentage points less likely to state they are willing to download the diagnostic

**Table 2. Treatment effects.**

| | Tracing App | | | | Diagnostic App | | | |
|---|---|---|---|---|---|---|---|---|
| | (1) | (2) | (3) | (4) | (5) | (6) | (7) | (8) |
| T1 (Facebook) | -0.000 | -0.004 | -0.004 | -0.004 | 0.002 | 0.001 | 0.000 | 0.001 |
| | (0.005) | (0.005) | (0.005) | (0.005) | (0.006) | (0.006) | (0.006) | (0.006) |
| T2 (GovOnlServ) | 0.008 | 0.006 | 0.006 | 0.006 | 0.024*** | 0.023*** | 0.023*** | 0.024*** |
| | (0.005) | (0.005) | (0.005) | (0.005) | (0.006) | (0.006) | (0.006) | (0.006) |
| T3 (DataPrivacy) | -0.029*** | -0.033*** | -0.032*** | -0.032*** | -0.042*** | -0.044*** | -0.044*** | -0.042*** |
| | (0.005) | (0.005) | (0.005) | (0.005) | (0.006) | (0.006) | (0.006) | (0.007) |
| Constant | 0.927*** | 0.925*** | 0.796*** | 0.937*** | 0.892*** | 0.901*** | 0.729*** | 0.915*** |
| | (0.003) | (0.012) | (0.105) | (0.030) | (0.004) | (0.013) | (0.113) | (0.035) |
| Observations | 22,776 | 21,251 | 21,193 | 21,070 | 22,724 | 21,194 | 21,137 | 21,017 |
| R-squared | 0.003 | 0.023 | 0.025 | 0.037 | 0.006 | 0.023 | 0.024 | 0.035 |
| Controls | No | Yes | Yes | Yes | No | Yes | Yes | Yes |
| Fixed Effects | No | No | State | Municipality | No | No | State | Municipality |
| T1 = T2 = T3 | 0.000 | 0.000 | 0.000 | 0.000 | 0.000 | 0.000 | 0.000 | 0.000 |
| T1 = T2 | 0.086 | 0.037 | 0.039 | 0.037 | 0.000 | 0.000 | 0.000 | 0.000 |
| T1 = T3 | 0.000 | 0.000 | 0.000 | 0.000 | 0.000 | 0.000 | 0.000 | 0.000 |
| T2 = T3 | 0.000 | 0.000 | 0.000 | 0.000 | 0.000 | 0.000 | 0.000 | 0.000 |

*Notes*: Each row shows the regression coefficients and the standard error in parenthesis corresponding to an OLS regression. Dependent variables take the value 0–1 according to the willingness of the respondent to download each application. Survey questions used for the construction of the dependent variables are available in S1 Appendix. Standard errors are robust.

*** p<0.01,

** p<0.05,

* p<0.1.

Controls include: sex, age, education, exposed to Covid, death to Covid, older than 65 at home, belief about infection probability, belief about hospitalization probability, attends party, visits family, risk inside evaluation, and others practice social distancing. Survey questions used for the construction of the control variables available in S1 Appendix. *Source*: Authors' calculations.

app, and 3 percentage points less likely to state willingness to download the contact tracing app, than those in the control group. Treatment T1, which refers to the usefulness of Facebook to keep in touch with others, has no effect. Interestingly, treatment T2—emphasizing government efforts to move procedures online—increases stated willingness to download the diagnostic app by about 2 percentage points, but does not impact willingness to download the contact tracing app, suggesting that indirectly emphasizing potential benefits may be a better way to motivate app adoption than talking about efforts to mitigate privacy risks. Equality-of-coefficients tests at the bottom of the table show that the treatment effect estimates mentioned in this paragraph are both statistically different from the control and statistically different from each other.

The results from the ordered logit model are shown in Fig 2. These estimates reveal that the negative treatment effect of the privacy issues treatment (T3) on the dichotomized stated likelihood to download either app reflects a reduction in the likelihood of answering "definitely yes" (about 6.5—7.5pp) alongside an increase in the likelihood of all other answer categories—with the biggest increase in the "I think so" category (about 4pp).

The coefficient estimates corresponding to the control variables (Fig 1) highlight baseline differences in stated willingness to download the apps across population subgroups. Older and more educated individuals are less likely to answer that they would download the apps. Women, those who were directly exposed or knew somebody who was exposed to (or died

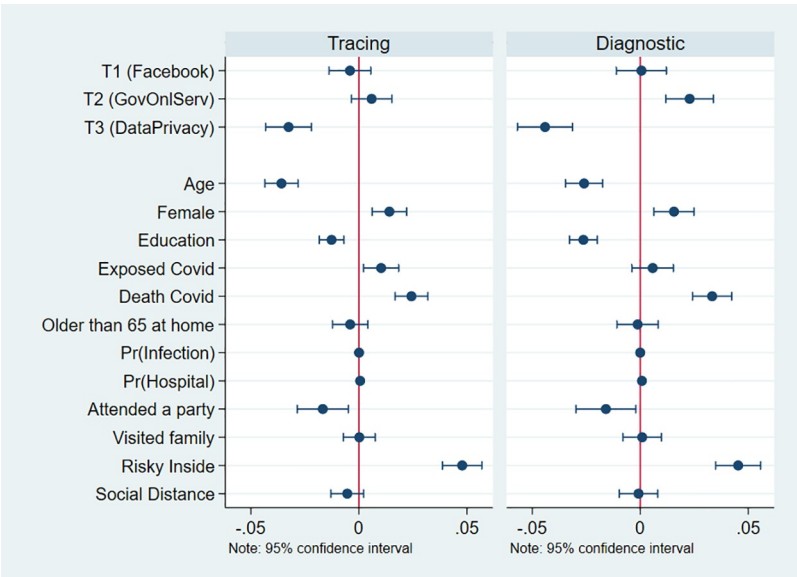

**Fig 1. Treatment effects and coefficient estimates.** This figure shows the Average Treatment Effects and the coefficients for the control variables. It corresponds to columns [2] and [6] in Table 2.

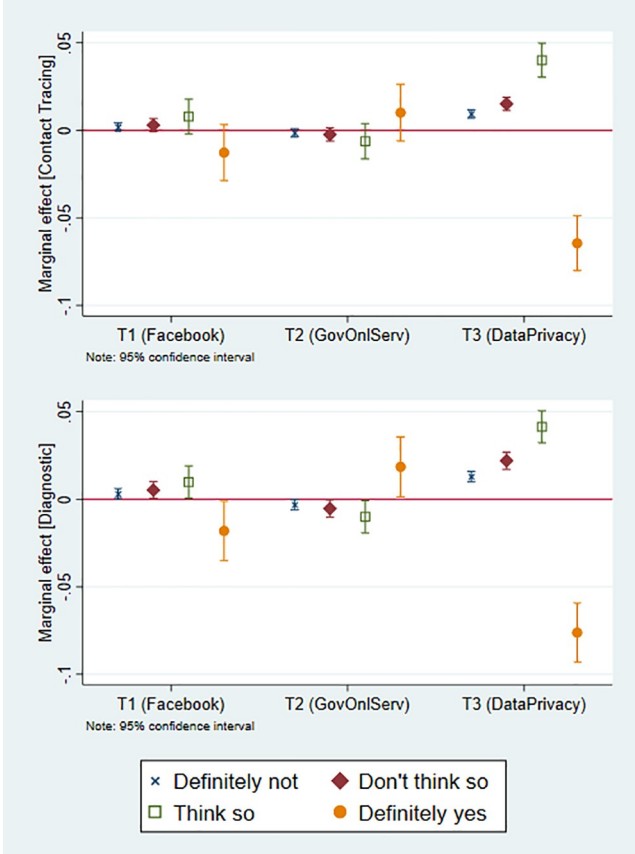

**Fig 2. Treatment effects—Ordered logit.** These figures show the change in probabilities associated to each treatment for the two dependent variables. Correspond to the margins of the coefficients in columns [1] and [4] in S1 Table.

because of) COVID-19, and those who thought it risky to carry out activities indoors with other people were all more likely to say that they would download the apps. In contrast, those who reported having attended a party in the previous 7 days were less likely to respond that they would download the apps. The direction of the effects of the control variables, elicited prior to the treatment assignment, is consistent with the idea that those with the greatest reason to be concerned about COVID-19 are more likely to download the apps, as might be expected. This is consistent with the idea that answers to the outcome questions about hypothetical download behavior have real substantive content.

## Robustness

In order to check whether something idiosyncratic about the wording of the data privacy treatment—rather than the fact that it draws attention to data privacy—is driving the results, we conducted a second survey experiment in a smaller sample of about 1,000 individuals. In that experiment, we added a fourth treatment. Individuals assigned to the new treatment received the following vignette:

> **T4**: *Ensuring citizen data privacy is of utmost importance for governments around the world, and Mexico is no exception. The data privacy of Mexicans is a priority for our government. Do you agree that protecting your privacy is a priority of the government?*
>
> [Yes/No]

This vignette aimed to emphasize, even more than T3, the actions that the government was taking to ensure data privacy, and to highlight that providing security was an explicit priority of the government.

Fig 3 summarizes the regression results (the full regression estimates are provided in S3 Table in the Supplementary Material). The results for treatments T1-T3 are very similar to those in the main analysis, serving as a replication exercise. Moreover, the results for the new

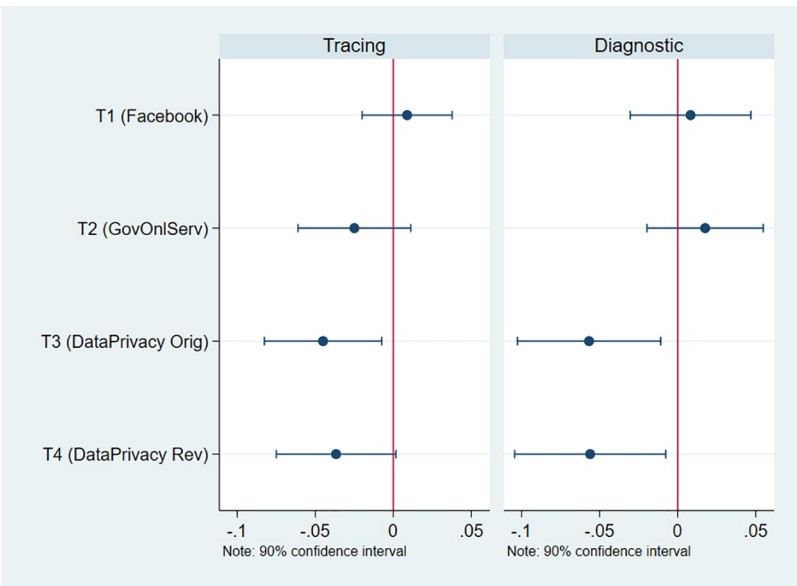

**Fig 3. Treatment effects—Sonora Sample.** This figure shows the treatment effects and coefficients for the two dependent variables. Correspond to columns [1] and [4] in S3 Table.

privacy treatment (T4) are virtually identical to those of the main privacy treatment (T3) for both of the willingness-to-download outcome questions. Respondents assigned to T3 and T4 were about 4—6pp less likely to state willingness to download either app. The fact that results are almost identical for T3 and T4 provides support for the idea that priming respondents about data privacy, regardless of the specific wording used, is the likely driver of the observed effect.

## Conclusion

During a pandemic, according to most experts, in order to control the spread of the virus, it is important to know who has the virus and who has been in contact with people who has it. That way, individuals infected can be isolated and receive adequate care. In order to achieve the levels of information necessary for the policy to be effective, governments have developed apps for self-diagnostic and for contact tracing. Individuals who suspect having the virus can seek professional help and isolate themselves to avoid potentially infecting others. Also, individuals can be informed when they have been in contact with somebody with a positive test. But for these apps to work, they have to be downloaded and used by a large fraction of the population. During the COVID-19 pandemic, governments have been relatively unsuccessful at getting citizens to download and use these apps. In order to increase take up, many resorted to highlighting their efforts to ensure data privacy. However, those very messages of reassurance may prime individuals to worry about data privacy and as a result reduce their willingness to download the apps.

In this paper, we bracket the issues that concern the ethics of the apps, their content, and the characteristics of the roll-out [9–11]. We concentrate instead on evaluating the informational content of the campaigns surrounding the roll-outs, and we present experimental evidence that stressing efforts to address concerns about data privacy may backfire. Mentioning privacy concerns appears to generate a 'knee-jerk' reaction against the download of the apps. This reaction is robust to two different wordings of the prime. Overall, our findings suggest that mentioning privacy concerns, instead of reassuring citizens, could convey the message that data privacy is something that citizens *should* be worrying about. It is also possible that the mere mention of "data privacy" might trigger a fear reaction. Discerning between these, and related, hypotheses about the precise mental processes at work is beyond the scope of this paper, but an interesting direction for future research. Additionally, future research may also want to test different messages regarding how the government might handle data privacy. For example, it is possible that messaging that explains how privacy is protected in a particular app may be persuasive, even if general statements about data privacy are not.

In contrast, avoiding mention of privacy concerns but focusing instead on the benefits of online government services increased the rate of stated willingness to download the apps. This positive effect may be due to the fact that this treatment highlights the government's positive record at making life easier for citizens by substituting online procedures for time consuming, in-person ones. It is also possible that this treatment indirectly emphasizes the benefits of using online services in general, thereby leading individuals to focus on the benefits of the apps rather than on their risks. The findings presented here may travel well to other related policy areas where safety is a concern, such as vaccination. More broadly, our results demonstrate the effectiveness of priming individuals as a means to influencing their (self-reported) intended behavior.

We have reason to believe that our results are likely to travel beyond Mexico's frontiers. According to data collected by [51], data privacy concerns among the Mexican public are very

similar to those in a sample of ten Latin American countries. The issue of data privacy is neither more nor less salient than in other countries in the region. The strength of legal protections of data privacy in Mexico is typical for the region, and above the regional mean according to V-DEM 2020 data; and the same is true for governmental cybersecurity capacity [52]. Moreover, trust in the government in Mexico is also about average (and above the median) for the region, according to the latest round of the Latinobarometer survey [53]. In sum, Mexicans are quite typical in terms of trust in government, concerns about data privacy, and the relevant legal environment.

## Supporting information

**S1 Fig. Facebook ads—Recruitment.** The figure shows a couple of examples of the ads used for recruitment. S2 Fig shows the different combinations of pictures used to construct these ads. These ads were designed by the project team and the IDB communications team.
(PDF)

**S2 Fig. Facebook ads—Set of pictures for the ads.** The figure shows the different pictures that were used to construct the set of ads used for recruitment.These ads were designed by the project team and the IDB communications team.
(PDF)

**S3 Fig. Distribution of responses—Control group.** This figure shows the distribution of responses to the questions regarding the download of the apps.
(PDF)

**S1 Table. Willingness to download the app—Ordered logistic regression.** The Table shows the regression coefficients corresponding to an ordered logit regression keeping the dependent variables in their original categorical values.
(PDF)

**S2 Table. Balance table—Sonora Sample.** This table shows descriptive statistics and balance among treatment assignment for each observable characteristic contained in the survey, for the Sonora Sample.
(PDF)

**S3 Table. Willingness to download the app—Sonora Sample.** This table presents the Average Treatment Effect for the Sonora Sample.
(PDF)

**S1 Appendix. Survey questions.** This document presents all the questions (in Spanish—original language of the survey—and English) used to construct dependent, treatment, as well as control variables.
(PDF)

**S1 File. Data set.** These 2 .dta files contain the underlying data set used to reach the conclusions drawn in this paper. One of the files corresponds to the overall sample for Mexico and the other for the more restricted sample for the state of Sonora.
(ZIP)

**S2 File. Regressions code.** This .do file allows readers to replicate the results of the paper using the S1 File.
(DO)

## Acknowledgments

We are thankful to the editor and two anonymous reviewers for their insights and suggestions, and to Susana Otálvaro for her assistance. The project would not have been possible without the help from Tom Sarrazin, the communications team at KIC, and the Government of the State of Guanajuato. All remaining errors are our own.

## Author Contributions

**Conceptualization:** Deborah Martínez, Cristina Parilli, Ana María Rojas, Carlos Scartascini, Alberto Simpser.

**Data curation:** Carlos Scartascini.

**Formal analysis:** Carlos Scartascini.

**Investigation:** Alberto Simpser.

**Methodology:** Carlos Scartascini, Alberto Simpser.

**Project administration:** Deborah Martínez, Cristina Parilli, Ana María Rojas.

**Supervision:** Deborah Martínez, Ana María Rojas, Carlos Scartascini.

**Writing – original draft:** Cristina Parilli, Carlos Scartascini, Alberto Simpser.

**Writing – review & editing:** Carlos Scartascini, Alberto Simpser.

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
