## [Decision Letter · Decision Letter 0]

4 May 2021

PONE-D-21-05671

Do you have COVID-19? How to increase the use of diagnostic and contact tracing apps

PLOS ONE

Dear Dr. Scartascini,

Thank you for submitting your manuscript to PLOS ONE. After careful consideration, we feel that it has merit but does not fully meet PLOS ONE’s publication criteria as it currently stands. Therefore, we invite you to submit a revised version of the manuscript that addresses the points raised during the review process, with particular focus on the points raised by Reviewer 1.

We look forward to receiving your revised manuscript.

Kind regards,

Noam Lupu

Academic Editor

PLOS ONE

Journal Requirements:

"Our survey experiment was part of a broader COVID-19-focused survey in Mexico, approved by the IRB of the Instituto Tecnológico Autónomo de México (ITAM) on July 1, 2020, under the name “Social and Behavioral Drivers of Individual Compliance with Preventive Measures during the COVID-19 Epidemic in Mexico” (ITAM’s IRB does not assign numbers to the projects it reviews)--memorandum letter of approval available upon request from the authors. We have obtained informed consent from all participants in this study.".   

Please provide additional details regarding participant consent. In the ethics statement in the Methods and online submission information, please ensure that you have specified what type you obtained (for instance, written or verbal, and if verbal, how it was documented and witnessed). If your study included minors, state whether you obtained consent from parents or guardians. If the need for consent was waived by the ethics committee, please include this information.

3. PLOS ONE has specific requirements for studies using personal data from third-party sources, including social media, blogs, other internet sources, and phone companies (https://journals.plos.org/plosone/s/submission-guidelines#loc-personal-data-from-third-party-sources). These requirements include confirming data are collected and used in accordance with the company or website’s Terms and Conditions, obtaining appropriate ethics or data protection body review, and ensuring appropriate consent from individuals whose data are used in research. In this case, please ensure that your Ethics statement is in compliance with guidelines, and that you have complied with the company's (i.e., Facebook's) Terms and Conditions, with appropriate permissions.

6. We note that Figures A1 and A2 in your submission contain copyrighted images. All PLOS content is published under the Creative Commons Attribution License (CC BY 4.0), which means that the manuscript, images, and Supporting Information files will be freely available online, and any third party is permitted to access, download, copy, distribute, and use these materials in any way, even commercially, with proper attribution. For more information, see our copyright guidelines: http://journals.plos.org/plosone/s/licenses-and-copyright.

1.         You may seek permission from the original copyright holder of Figures A1 and A2 to publish the content specifically under the CC BY 4.0 license.

7. Thank you for providing the following Funding Statement: 

[The Inter-American Development Bank and ITAM provided support in the form of salaries for all the authors, but did not have any additional role in the study design,data collection and analysis, decision to publish, or preparation of the manuscript.].

We note that one or more of the authors is affiliated with the funding organization, indicating the funder may have had some role in the design, data collection, analysis or preparation of your manuscript for publication; in other words, the funder played an indirect role through the participation of the co-authors.

If the funding organization did not play a role in the study design, data collection and analysis, decision to publish, or preparation of the manuscript and only provided financial support in the form of authors' salaries and/or research materials, please review your statements relating to the author contributions, and ensure you have specifically and accurately indicated the role(s) that these authors had in your study in the Author Contributions section of the online submission form. Please make any necessary amendments directly within this section of the online submission form.  Please also update your Funding Statement to include the following statement: “The funder provided support in the form of salaries for authors [insert relevant initials], but did not have any additional role in the study design, data collection and analysis, decision to publish, or preparation of the manuscript. The specific roles of these authors are articulated in the ‘author contributions’ section.”

If the funding organization did have an additional role, please state and explain that role within your Funding Statement.

Please also provide an updated Competing Interests Statement declaring this commercial affiliation along with any other relevant declarations relating to employment, consultancy, patents, products in development, or marketed products, etc.  

Reviewers' comments:

Reviewer's Responses to Questions

**Comments to the Author**

1. Is the manuscript technically sound, and do the data support the conclusions?

Reviewer #1: Partly

Reviewer #2: Yes

2. Has the statistical analysis been performed appropriately and rigorously? 

Reviewer #1: Yes

Reviewer #2: N/A

3. Have the authors made all data underlying the findings in their manuscript fully available?

Reviewer #1: Yes

Reviewer #2: Yes

4. Is the manuscript presented in an intelligible fashion and written in standard English?

Reviewer #1: Yes

Reviewer #2: Yes

5. Review Comments to the Author

Reviewer #1: This study investigates the important issue of uptake of COVID-19 related apps intended for self-diagnosis and digital contact tracing. Using a survey experiment among a large sample of Facebook users in Mexico, it asks whether emphasizing data privacy, as many governments have done, is a useful messaging strategy or whether it might backfire by making privacy concerns more salient to the public. The authors find evidence in support of this idea: survey respondents primed with a message about data privacy were less likely to state that they would download hypothetical diagnostic and contact tracing apps, compared to a control group and to groups that received other messages.

This is an important contribution as governments and app developers consider why the uptake of such apps has been low and the study is generally well-executed. However, I have several suggestions for improvement, mostly related to the interpretation of certain results, scope conditions, and interpretation and implications. In the attached comments I discuss each of these, listed roughly in order of importance and list some additional more minor comments.

Reviewer #2: The paper presents original results about the way governments' messages regarding privacy safeguards, contrary to their intended aim, may actually trigger concerns and undermine citizens' willingness to download COVID-19 symptom checkin and contact tracing apps. I have read the article with great interest. I think the paper is solid and well-written.

I have only one suggestion that I hope the authors will be willing to take into account. The discussion section could be expanded and engage with two bodies of literature. On the one hand the empirical literature on public attitudes towards COVID-19 apps in other geographical contexts. For instance:

Saw, Y.E., Tan, E.Y.Q., Liu, J.S. and Liu, J.C., 2021. Predicting public uptake of digital contact tracing during the covid-19 pandemic: results from a nationwide survey in Singapore. Journal of medical Internet research, 23(2), p.e24730.

Zimmermann, B.M., Fiske, A., Prainsack, B., Hangel, N., McLennan, S. and Buyx, A., 2021. Early perceptions of COVID-19 contact tracing apps in German-speaking countries: comparative mixed methods study. Journal of medical Internet research, 23(2), p.e25525.

On the other the literature (mostly in the field of ethics) stressing privacy as the main (ethical) concern, especially in the case of contact tracing apps. For instance

Mello, M.M. and Wang, C.J., 2020. Ethics and governance for digital disease surveillance. Science, 368(6494), pp.951-954.)

Parker, M.J., Fraser, C., Abeler-Dörner, L. and Bonsall, D., 2020. Ethics of instantaneous contact tracing using mobile phone apps in the control of the COVID-19 pandemic. Journal of Medical Ethics, 46(7), pp.427-431.

This literature focuses mostly on privacy and autonomy (voluntariness) in their ethical analysis.

Some other scholars in ethics instead have highlighted the narrow focus on privacy, see:

Martinez‐Martin, N., Wieten, S., Magnus, D. and Cho, M.K., 2020. Digital contact tracing, privacy, and public health. Hastings Center Report, 50(3), pp.43-46.

It would be nice if you could situate your results in this broader context.

On last observation. Your point (e.g. in the last paragraph) that focusing on benefit increases the willingness to download the app, is a bit undersold. Discussing it in light of the above mentioned debates would do it more justice. But other than that, at a more general level, this kind of consideration reminded me of a - now dated - paper by bioethicist Bruce Jennings on Public Health and Civic Republicanism (available here: https://www.researchgate.net/publication/262724398_Public_Health_and_Civic_Republicanism). In this paper Jennings argues that exclusive emphasis on human rights and individual interest will not do the job of providing a credible public justification to public health measures (page 57). Jennings appeals instead to messages centred around civic virtues (i.e. doing it for the good of all). I thought you could take this into account as a theoretical backdrop towards the end of your discussion.

6. PLOS authors have the option to publish the peer review history of their article (what does this mean?). If published, this will include your full peer review and any attached files.

Reviewer #1: No

Reviewer #2: No

---

## [Author Response · Author response to Decision Letter 0]

27 May 2021

Responses to all comments raised by the editor and the reviewers have been attached in a separate file

---

## [Editor Report · Decision Letter 1]

7 Jun 2021

Do you have COVID-19? How to increase the use of diagnostic and contact tracing apps

PONE-D-21-05671R1

Dear Dr. Scartascini,

We’re pleased to inform you that your manuscript has been judged scientifically suitable for publication and will be formally accepted for publication once it meets all outstanding technical requirements.

Kind regards,

Noam Lupu

Academic Editor

PLOS ONE
---

## [Editor Report · Acceptance letter]

21 Jul 2021

PONE-D-21-05671R1 

Do you have COVID-19? How to increase the use of diagnostic and contact tracing apps 

Dear Dr. Scartascini:

I'm pleased to inform you that your manuscript has been deemed suitable for publication in PLOS ONE. Congratulations! Your manuscript is now with our production department. 

Kind regards, 

on behalf of

Dr. Noam Lupu 

Academic Editor

PLOS ONE